# Regenerating end-of-life membranes for enhanced sustainability and unexpected performance

Chenxin Tian [1,3], Jiansuxuan Chen[1,3], Zhiwei Qiu[1], Ruobin Dai [1] ✉, Shihong Lin [2] & Zhiwei Wang [1] ✉

Membrane technologies are widely adopted in water purification, gas separation, resource recovery and chemical production. However, membranes eventually reach their end-of-life (EOL) due to structural degradation and irrecoverable fouling, leading to incineration or landfilling which contradicts the sustainability and circular economy principles. Here we present a strategy to regenerate EOL membranes through dissolution in organic solvent followed by re-casting. The regenerated membrane exhibits more than fivefold higher water permeance with improved pollutant rejection compared to the EOL membrane, and even outperforms membrane fabricated from pristine polymer powders. This enhancement is attributed to the integration of residual foulants as pore-forming agents and hydrophilic additives. Moreover, the reduced entanglement density of EOL membrane improves its compatibility with solvent and foulants, enabling the formation of a dense separation layer in the regenerated membrane. This strategy achieves 38.4% lower $CO_2$-eq emissions and 75.7% cost reduction, advancing sustainability and circularity of the membrane industry.

Membrane technologies have been widely adopted in water purification, gas separation, resource recovery and chemical production[1–5]. In the membrane industry, membranes typically follow a linear life-cycle, including production, usage and disposal stages[6]. The production of polymeric membranes predominantly relies on non-biodegradable, petroleum-based polymer materials[7,8]. The use of these polymer materials is accompanied by the extraction, transportation, and refining of non-renewable crude oil, which inevitably contributes to marine and air pollution, as well as an increased carbon footprint[9].

Current membranes have a finite lifespan ranging between 3 to 7 years, due to the accumulation of irrecoverable fouling and possible structural damage over prolonged service periods (Fig. 1a)[10,11]. Subsequently, end-of-life (EOL) membranes are disposed through landfilling or incineration, imposing negative impacts on the environment and climate (Fig. 1b)[12,13]. Moreover, the need to replace EOL

membranes places a substantial economic burden on membrane users. To reduce the environmental impacts associated with petroleum usage and EOL membrane disposal, a transition from linear life cycle to circular model is urgently needed in the membrane industry[14–16].

One of the most common methods for preparing polymeric membranes is phase inversion, which involves the controlled transformation of an initially homogeneous polymer casting solution into a three-dimensional polymer network[17,18]. During this process, the polymer transitions from a discontinuous dispersion to a continuous solid phase via a physical phase change without any chemical reaction[19,20]. Therefore, EOL membranes can potentially be repurposed as polymer materials for membrane regeneration through phase inversion, thereby establishing circularity within the membrane industry.

[1]State Key Laboratory of Water Pollution Control and Green Resource Recycling, Shanghai Institute of Pollution Control and Ecological Security, School of Environmental Science and Engineering, Tongji University, Shanghai, China. [2]Department of Civil and Environmental Engineering, Vanderbilt University, Nashville, Tennessee, USA. [3]These authors contributed equally: Chenxin Tian, Jiansuxuan Chen. ✉e-mail: dairuobin@tongji.edu.cn; zwwang@tongji.edu.cn

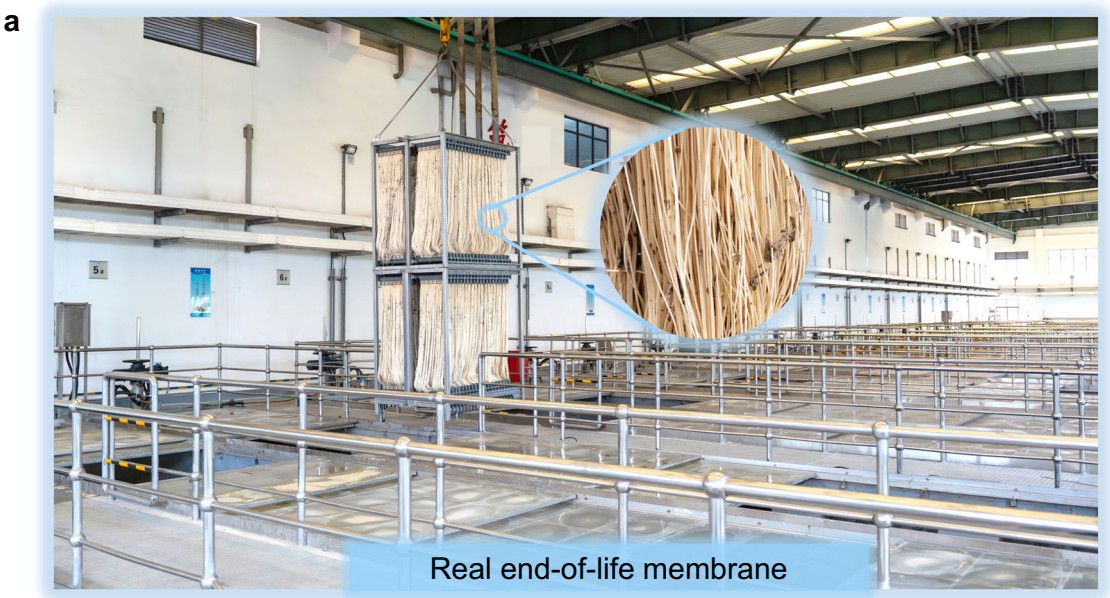

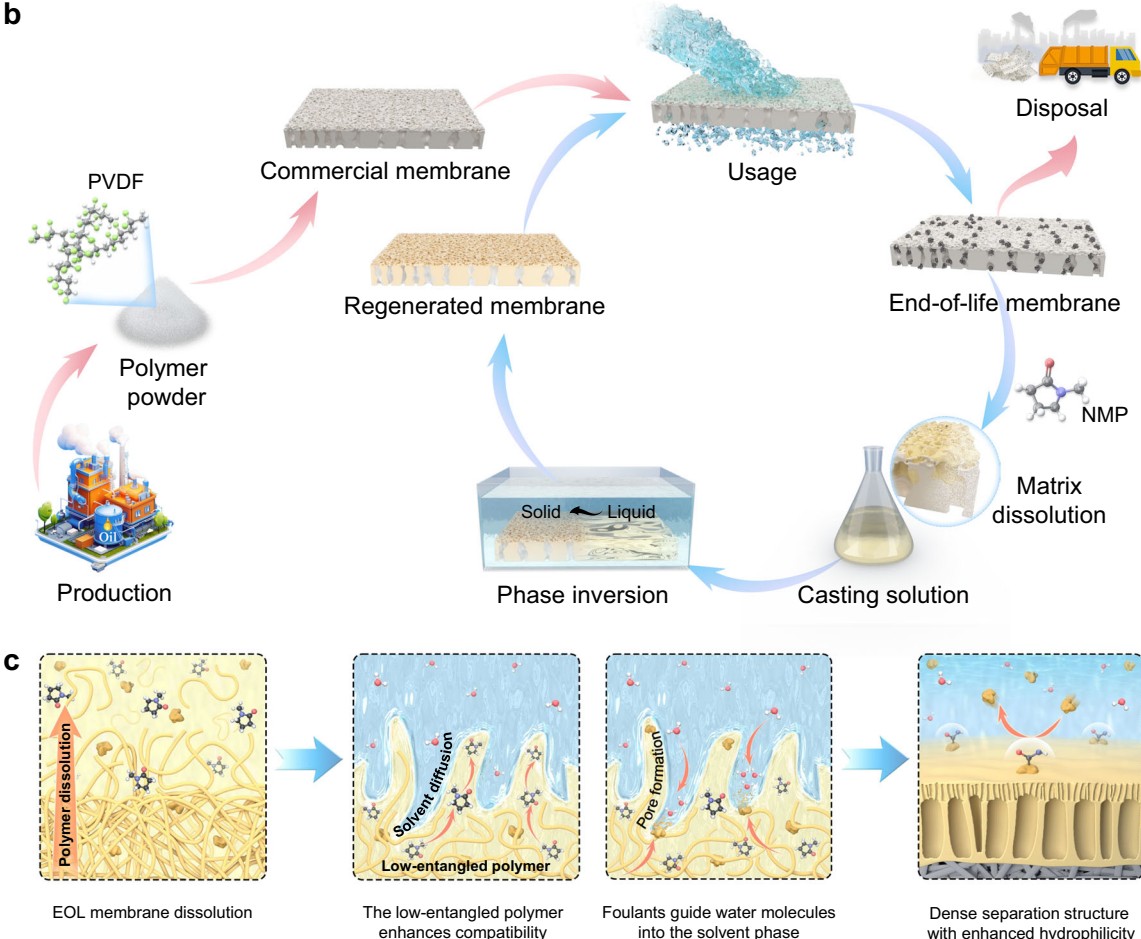

**Fig. 1 | The EOL membrane and schematic of membrane regeneration strategy based on solvent-driven membrane regeneration. a**, The EOL hollow fiber polyvinylidene fluoride (PVDF) membranes from a full-scale membrane bioreactor. **b**, Sustainable membrane preparation with EOL membranes. **c**, Schematic of membrane regeneration for creation of the dense separation structure and hydrophilic surface of the regenerated membrane. The low-entangled polymer facilitates compatibility enhancement of casting solution, and the presence of foulants enhances the hydrophilicity of the membrane surface.

Two potential challenges with regenerating EOL membranes are (1) the change of polymer properties due to prolonged exposure of the membrane to cleaning agents in operation; and (2) the presence of foulants trapped in the pores of EOL membranes that cannot be readily removed. Due to these two factors, the reused polymer from EOL membranes is expected to be substantially different from the pristine polymer used to fabricate the new commercial membranes. The impacts of these two factors on the quality and performance of the regenerated membranes are critical to the practical potential of EOL membrane regeneration but have not been systematically investigated. It is thus the goal of this study to examine the impacts of residual foulants and change in polymer properties on the characteristics and performance of regenerated EOL membranes.

Herein, we report a method of regenerating EOL membranes using solvent dissolution followed by re-casting (Fig. 1b). The regenerated membrane demonstrated an enhanced water permeance and improved pollutant rejection not only compared to the EOL membrane, but surprisingly, even to pristine membrane prepared using commercial polyvinylidene fluoride powders. To understand this

unexpected result, we further analyze the important roles of foulant integration and polymer entanglement change in enhancing the performance of the regenerated membrane (Fig. 1c). Moreover, we conduct life cycle assessment and economic analysis to show both environmental and economic advantages of the membrane regeneration strategy. The membrane regeneration method opens an avenue to enhance the sustainability of the membrane industry.

## Results

### Enhanced membrane performance after regeneration

The EOL hollow fiber PVDF membranes with a symmetrical structure (Fig. 2a) were covered by substantial amount of foulants (Fig. 2b) when retrieved from a full-scale membrane bioreactor (MBR). The regenerated EOL membranes (labeled as EOL-R, see Methods Section for details of the regeneration process) exhibits a pale-yellow appearance with a skin top layer and finger-like pores that are typical of those forming from phase inversion (Fig. 2c–e). EOL-R membrane has a high water permeance ($233.0 \pm 12.6$ L m$^{-2}$ h$^{-1}$ bar$^{-1}$) and bovine serum albumin (BSA) rejection ($80.4 \pm 2.5\%$), dwarfing the EOL membrane with a

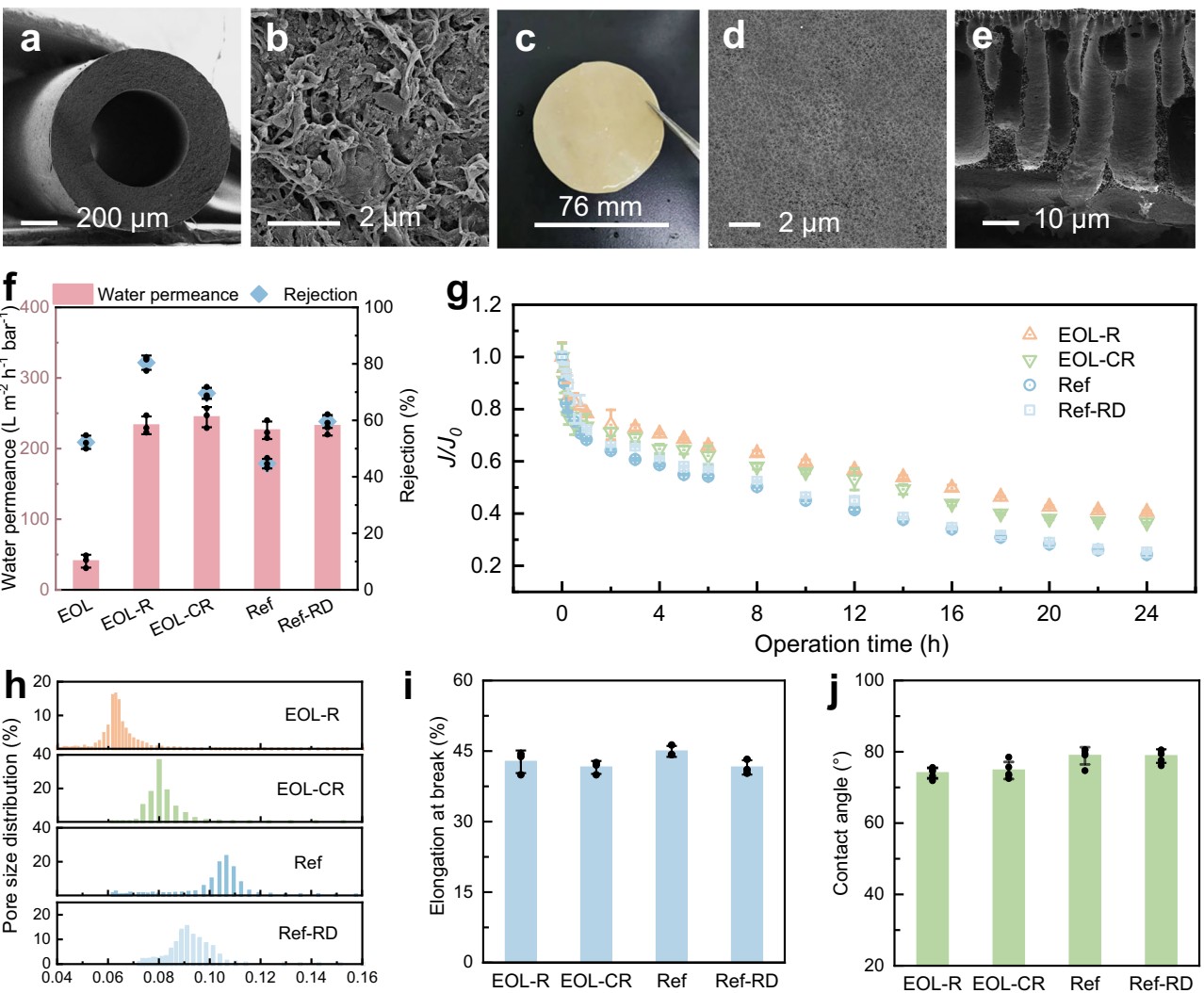

**Fig. 2 | Performance and characterization of the regenerated membrane.** Characterization of the EOL membrane: (**a**) Cross-sectional SEM image. **b** SEM image of membrane surface. Characterization of the regenerated membrane: (**c**) Optical image. **d** SEM image of membrane surface. **e** Cross-sectional SEM image. **f** Water permeance and BSA rejection of the EOL, EOL-R, EOL-CR, Ref and Ref-RD membranes. EOL-R membrane was prepared by regenerating EOL membrane. EOL-CR membrane was prepared by regenerating cleaned EOL membrane (foulants

were removed). The Ref membrane was prepared using a commercial polymer. Ref-RD membrane was prepared using Ref membrane. Fouling-resistance performance of membranes: (**g**) variations of $J/J_0$ with operation time for membranes with 200 mg L$^{-1}$ HA aqueous solution at neutral pH under 0.1 MPa. Characterization of membranes: (**h**) Pore size distribution. **i** Elongation at break. Error bars in (**f, g, i**) represent the s.d. ($n = 3$), and data are presented as mean values ± s.d. **j** Contact angle ($n = 5$). Source data are provided as a Source Data file.

deteriorated water permeance ($40.5 \pm 9.1$ L m$^{-2}$ h$^{-1}$ bar$^{-1}$) and a low BSA rejection of $52.2 \pm 2.4\%$ (Fig. 2f).

To investigate the impact of foulants, EOL membranes were thoroughly cleaned to remove foulants prior to regeneration (see Methods Section and Supplementary Fig. 1), and the resulting membrane was referred to as the EOL-CR membrane. Compared to the EOL-R membrane, the EOL-CR membrane exhibited a slightly higher water permeance ($244.5 \pm 14.2$ L m$^{-2}$ h$^{-1}$ bar$^{-1}$) but a lower BSA rejection ($69.6 \pm 2.0\%$ vs. $80.4 \pm 2.5\%$) (Fig. 2f). The improved rejection performance of the EOL-R membrane could be attributed to the presence of residual foulants, which may play a beneficial role in tuning membrane structure and improving separation performance during regeneration. Reference membrane (labeled as Ref membrane) was also synthesized using pristine polymer powder, following the exact method for fabricating the EOL-R and EOL-CR membranes. The Ref membranes showed a lower water permeance ($225.9 \pm 12.4$ L m$^{-2}$ h$^{-1}$ bar$^{-1}$) and a substantially lower BSA rejection rate ($44.7 \pm 1.7\%$) compared to the EOL-R membrane. Then the Ref membrane was dissolved using solvent, and a redissolved membrane (labeled as Ref-RD membrane) was prepared through an identical phase inversion process. The Ref-RD membrane exhibited a similar water permeance ($232.1 \pm 13.6$ L m$^{-2}$ h$^{-1}$ bar$^{-1}$) but a slightly higher pollutant rejection ($59.6 \pm 2.3\%$) compared to the Ref membrane. However, both membranes demonstrated significantly lower rejection performance compared to the EOL-R membrane, underscoring the potential role of residual foulants in enhancing separation efficiency.

Fouling resistance is crucial for the long-term durability and efficiency of membranes. The fouling-resistance performance of regenerated membranes was evaluated by conducting fouling experiments under constant pressure, using three model foulants: humic acid (HA), sodium alginate (SA) and BSA (Fig. 2g and Supplementary Fig. 2). During the first 0.5 h of filtration, the permeate flux showed a rapid decline, followed by a slower decrease. Interestingly, both regenerated membranes (EOL-R and EOL-CR) exhibited a similar flux decline behavior, which was consistently slower than that observed for the Ref and Ref-RD membranes. After 24 h of filtration with the HA solution, the $J/J_0$ of EOL-R and EOL-CR decreased to 0.41 and 0.37, respectively, which were significantly higher than that of Ref and Ref-RD membranes (0.24 and 0.25) (Fig. 2g). Similar results were observed in the filtration experiments using BSA and SA solutions (Supplementary Fig. 2), further confirming that the EOL-R membrane, prepared with EOL membrane containing a considerable amount of foulants, exhibited the best fouling resistance.

## Smaller pores and better wettability in regenerated membrane

The regenerated membranes have smaller pores (0.064 μm for EOL-R and 0.082 μm for EOL-CR) as compared to the Ref and Ref-RD membranes (0.107 μm and 0.088 μm) (Fig. 2h), contributing to the superior rejection performance of the EOL-R and EOL-CR membranes. Besides, the characterization of cross-sectional morphology indicates that EOL-R and EOL-CR membranes have a denser structure of the separation layer compared to the Ref membrane (Fig. 2e and Supplementary Fig. 3a, b). Despite being prepared with EOL membrane, the regenerated membrane displayed a crystalline structure similar to that of the Ref membrane (Supplementary Fig. 3c). In addition, regenerated membranes showed enhanced thermal stability and comparable mechanical stability, as evidenced by a higher melting temperature ($T_m$) (138.3 °C) than that of the Ref membrane (135.6 °C) (Supplementary Fig. 3d), as well as a nearly similar elongation at break (Fig. 2i).

In addition, the regenerated membranes displayed enhanced hydrophilicity as evidenced by a smaller water contact angle (Fig. 2j). The improved surface hydrophilicity is likely attributable to the presence of the incorporated foulants, which is supported by the lower F $1s$ and higher O $1s$ content observed in the regenerated membrane as compared to the Ref membrane (Supplementary Table 1). Moreover,

the regenerated membrane demonstrated a smoother surface ($R_a = 41.5 \pm 9.7$ nm for EOL-R and $38.2 \pm 5.6$ nm for EOL-CR) compared to the Ref membrane ($R_a = 64.6 \pm 6.9$ nm) (Supplementary Fig. 3e). These changes in surface characteristics play important roles in enhancing the fouling resistance of the regenerated membrane. Furthermore, we have additionally collected EOL membranes from two other application scenarios for membrane regeneration (Supplementary Method 1 and Fig. 4). The regenerated membranes consistently exhibited significantly improved performance compared to the original EOL membranes, confirming the broad applicability of our regeneration strategy.

## Role of residual foulant in membrane regeneration

To elucidate the impacts of integrated foulants on regenerated membranes, BSA (as model organic foulant) and silicon dioxide (SiO$_2$, as model inorganic foulant) were individually blended into EOL-CR casting solution as membrane additives, which have been detected in EOL membranes[21]. The resulting membranes, containing 0, 3, 6, and 9 wt% of foulant (foulant/PVDF, $w/w$), were labeled as EOL-CR-foulant-0, -1, -2, and -3, respectively. The incorporation of BSA and SiO$_2$ enhanced BSA rejection (~90%) while reducing water permeance (Fig. 3a). These changes in membrane performance align with the differences observed between EOL-R and EOL-CR membranes, suggesting that the improved rejection and slightly lower permeance of the EOL-R membrane indeed result from the presence of foulants. As the foulant dosage increased from 0 wt% to 9 wt%, the average pore size of EOL-CR-BSA and EOL-CR-SiO$_2$ membranes slightly decreased (from 0.082 μm to 0.066 μm and 0.071 μm, respectively) (Fig. 3b). This trend indicates that foulants act as additional diffusion sites for water molecules during the phase inversion process, thereby facilitating the formation of smaller pores.

Furthermore, foulant incorporation enhanced the surface hydrophilicity of membranes, especially for EOL-CR membranes with SiO$_2$ integrated (Fig. 3c). FTIR analysis confirmed the presence of hydrophilic functional groups: amide (1660 cm$^{-1}$ and 1550 cm$^{-1}$) in EOL-CR-BSA and Si-O-Si bond (1110 cm$^{-1}$) in EOL-CR-SiO$_2$ membrane (Supplementary Fig. 5)[22,23]. These findings suggest that foulants function not only as pore-forming agents during membrane fabrication but also remain partially embedded in the membrane matrix to enhance hydrophilicity. Given the widespread occurrence of these foulant components in diverse wastewater treatment applications[24–26], their positive contribution to the performance of regenerated membranes is likely to have universal applicability.

Similarly, we added foulants into the Ref membrane, $i.e.$, incorporating BSA and SiO$_2$ with pristine polymer powder, to prepare Ref-foulant membranes. This incorporation also led to a decrease in pore size (Supplementary Fig. 6) and enhancement in surface hydrophilicity (Supplementary Fig. 7), contributing to a slight improvement in BSA rejection for Ref-foulant membranes (Fig. 3d). Nevertheless, compared to Ref-foulant membranes, EOL-CR-foulant membranes demonstrated concurrent enhancements in both water permeance and BSA rejection (Fig. 3e). This phenomenon suggests that utilizing EOL membranes as polymer materials could amplify the beneficial effects of foulant integration on membrane separation performance.

The cross-sectional morphology of EOL-CR-foulant membranes appeared more structured than that of Ref-foulant membranes (Supplementary Fig. 8 and Fig. 9), likely due to the superior compatibility of EOL membranes with foulants. Notably, the cross-sections of Ref-foulant membranes exhibited a higher accumulation of foulants within the pores compared to EOL-CR-foulant membranes, as indicated by the increased content of O and Si elements (Fig. 3f, g). Besides, SiO$_2$ particles were observed to block more pores in the Ref-SiO$_2$-3 membrane than in the EOL-CR-SiO$_2$-3 membrane (Supplementary Fig. 10). These findings suggest that utilizing EOL membranes as polymer materials could promote better integration of foulants into the

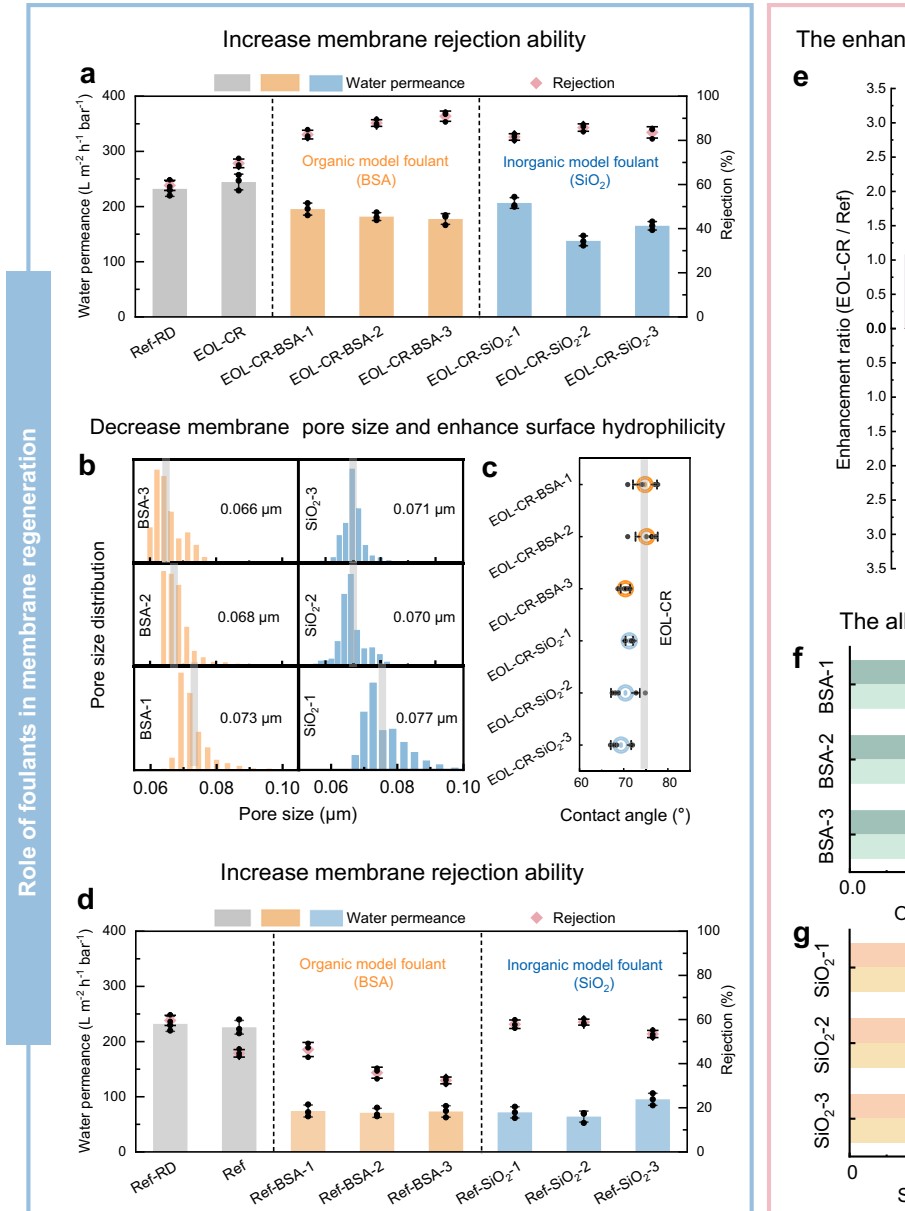

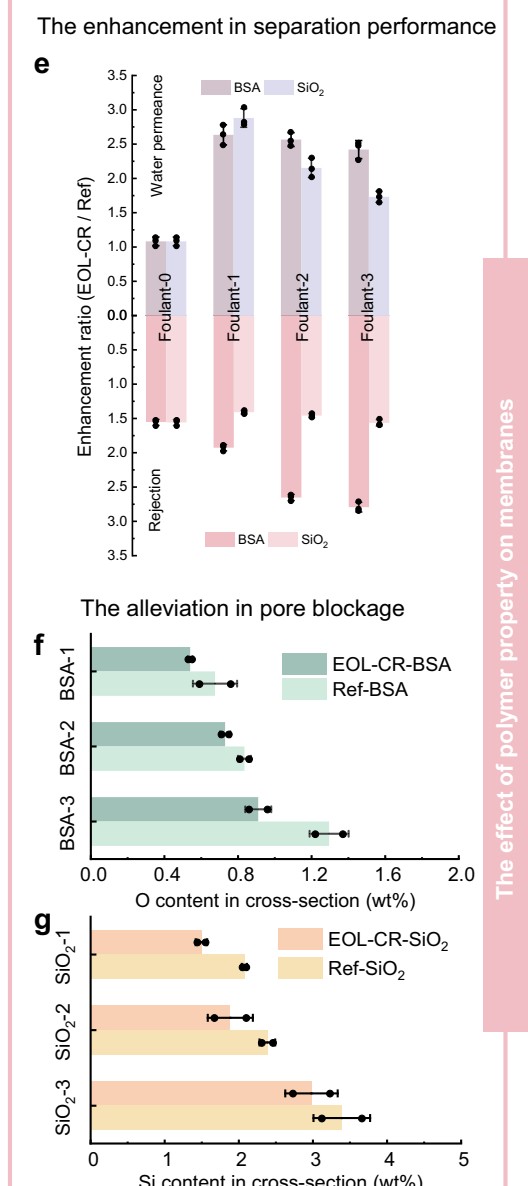

**Fig. 3 | The effect of foulant incorporation on membrane performance.** The membranes blended with foulant at 0, 3, 6, and 9 wt% was labeled as EOL-CR-foulant-0, 1, 2, 3, respectively. The performance and characterization of EOL-CR-foulant membranes: (**a**) The water permeance and rejection rate. **b** Pore size distribution. The gray bar represents the average pore size of the membrane. **c** Water contact angle ($n = 5$). The gray bar represents the water contact angle of the EOL-CR membrane. **d** The water permeance and rejection rate of Ref-foulant membranes. **e** Performance enhancement ratio of EOL-CR-foulant membranes compared to Ref-foulant membranes: water permeance and rejection rate. The element content in cross section of foulant-incorporated membranes: (**f**) O element content in cross section of EOL-CR-BSA and Ref-BSA membranes. **g** Si element content in cross-section of EOL-CR-SiO$_2$ and Ref-SiO$_2$ membranes. Error bars in (**a**, **d**, **e**, **f**, **g**) represent the s.d. ($n = 3$), and data are presented as mean values ± s.d. Source data are provided as a Source Data file.

membrane matrix rather than just retention within membrane pores, facilitating the formation of more structured cross-section and the enhancement of membrane separation performance. For EOL membranes, substantial foulants accumulated on their surface or in membrane pores. Comparatively, foulants in EOL-R membranes have migrated into the membrane matrix, contributing to the substantial enhancement of water permeance.

**Unraveling polymer property difference among membranes**
Unlike commercial polymer powders, EOL membranes have experienced phase inversion during membrane fabrication and have been subjected to prolonged exposure to cleaning agents during operation.

To investigate the difference in polymer entanglement between the EOL membrane and commercial powder, the plateau moduli ($G^0_N$) of EOL-R, EOL-CR, Ref and Ref-RD casting solutions were determined by measuring changes in loss modulus ($G''$) as a function of frequency (Fig. 4a). The maximum loss modulus ($G''_{max}$) was observed between 400 to 500 rad/s, and $G^0_N$ was calculated according to Eq. (1). Both EOL-R and EOL-CR casting solutions exhibited lower $G^0_N$ values compared to that of the Ref casting solution (Fig. 4b). Given that entanglement density ($V_e$) is directly proportional to the plateau modulus ($V_e \approx G^0_N/k_bT$, where $k_b$ is the Boltzmann constant and $T$ is the temperature)[27], the solution prepared by dissolving the EOL membrane demonstrated a significantly lower entanglement density than

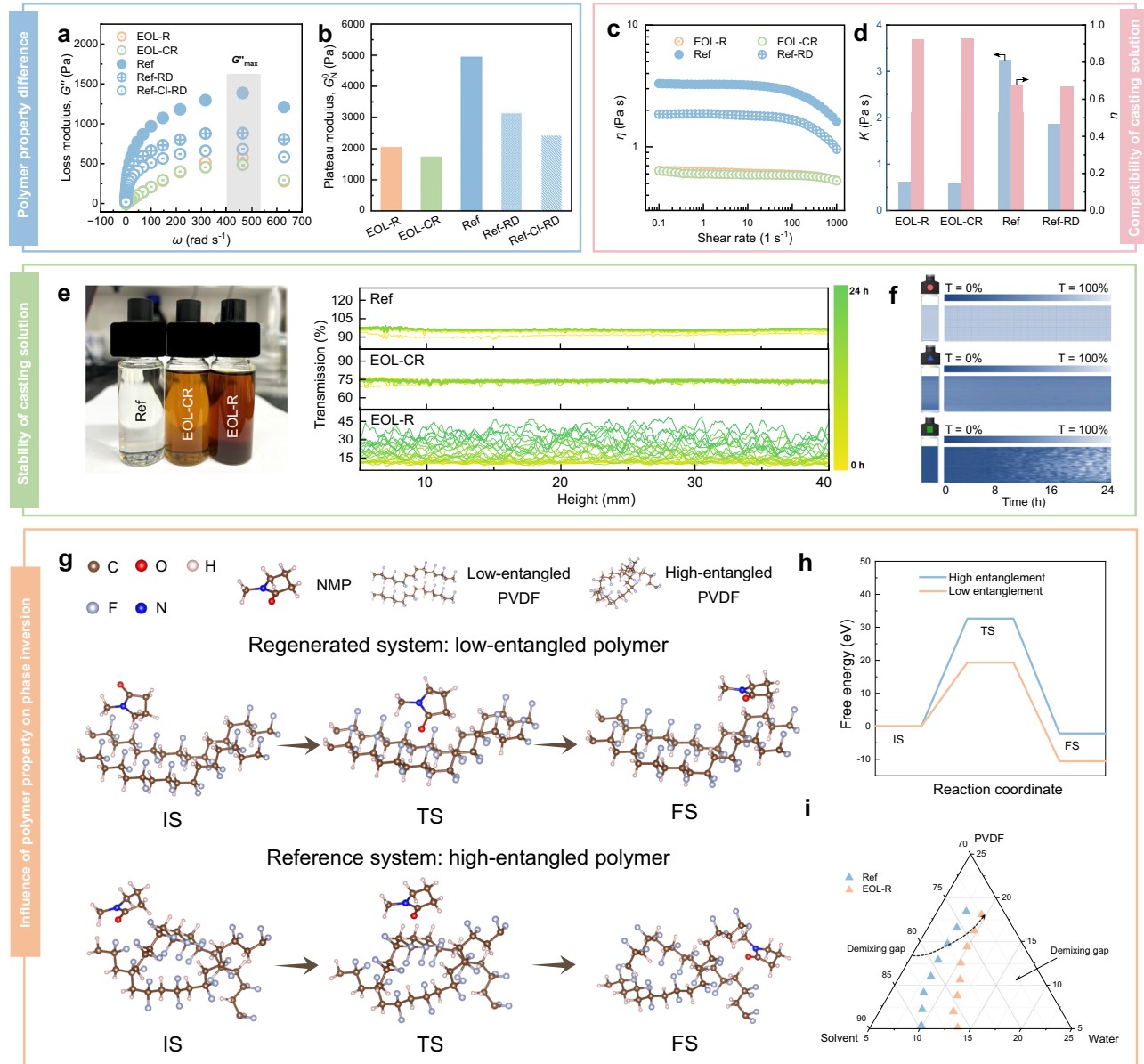

**Fig. 4 | Mechanisms of regenerated membrane formation.** The determination of plateau modulus of various casting solutions: (**a**) Loss modulus as a function of angular velocity. The gray shaded region refers to the maximum of loss modulus ($G''_{max}$) of membranes. **b** Plateau modulus. The rheology characterization of casting solutions: (**c**) Viscosity as a function of shear rate. **d** $K$ and $n$ values. **e** Optical image and transmission intensity profiles along the vertical direction of the Ref, EOL-CR, and EOL-R casting solutions. **f** Reconstructed images of three casting solutions from Turbiscan Tower. The Left vessel in each reconstructed diagram shows the initial existing status of the casting solution sample and the height-time plot shows the changes of existing status throughout the measurement period. Density functional theory calculations: (**g**) Illustration of diffusion of NMP molecules near low and high entanglement of PVDF chains in initial state (IS), transition state (TS) and final state (FS). **h** Migration energy barrier of NMP. **i** A schematic phase diagram of the PVDF/NMP/water ternary system for regenerated and reference systems. The two cloud point curves were determined through titration experiments. Source data are provided as a Source Data file.

the solution prepared from commercial powders[28,29]. This reduction is likely due to the effects of phase inversion and cleaning processes.

Specifically, Ref-RD casting solution showed a lower $G^0_N$ value (3135.1 Pa) compared to the Ref casting solution (4933.1 Pa) (Fig. 4b), indicating that the phase inversion process contributed to the decrease in entanglement density of polymer. To explore the effect of prolonged oxidative membrane cleaning on the polymer properties of the EOL membrane, we further performed an experiment to simulate such an effect on pristine polymer powders. The Ref membranes were exposed to a high concentration of sodium hypochlorite (NaClO) to mimic the long-term effects of cleaning agents (Supplementary Method 2), and then were re-dissolved to prepare casting solutions labeled as Ref-Cl-RD casting solution. The Ref-Cl-RD casting solution

exhibited an even lower $G^0_N$ value (2422.0Pa) compared to the Ref-RD casting solution, indicating that prolonged exposure to oxidative membrane cleaning also reduced the entanglement density of the polymer (Fig. 4a, b).

## How does the polymer property affect membrane performance?

Polymers with lower entanglement tend to fully extend and uniformly disperse in the solvent, thereby enhancing the stability and compatibility of the casting solution during membrane regeneration. The compatibility and stability of the casting solution, along with the mass transfer kinetics during the phase inversion process, collectively determine the microstructure and separation performance of the resulting membrane[30,31]. Therefore, we further evaluated the impact of

polymer entanglement on compatibility and stability of casting solutions through rheology characterization and multiple light scattering spectroscopy (MLiSSP) characterization, respectively[32,33].

The compatibility between the EOL membrane and solvent was better compared to the pristine polymer powder, as verified by the notably lower viscosity of the EOL-R and EOL-CR casting solutions compared to that of the Ref and Ref-RD casting solutions (Fig. 4c). The casting solution is essentially a macromolecular polymer dispersion whose viscosity typically exhibits shear-rate dependency due to changes in polymer conformation[34]. Above a critical shear rate, the viscosity of the Ref and Ref-RD casting solutions decreased with increasing shear rate (Fig. 4c), which can be attributed to the coiled and entangled PVDF chains becoming disentangled and eventually straightening out at high shear rates[35]. In contrast, the viscosities of EOL-R and EOL-CR casting solutions were minimally affected by shear rate (i.e., the PVDF chains possessed consistently low entanglement), displaying behavior indicative of ideal Newtonian fluidity (approaching $n = 1$) (Fig. 4d). Furthermore, the EOL-R and EOL-CR casting solutions exhibited a significantly lower $K$ value than the Ref and Ref-RD casting solutions, indicating their superior polymer-solvent compatibility[34].

The stability of EOL-R, EOL-CR and Ref casting solutions was evaluated by monitoring the transmission intensities over time with multiple light scattering spectroscopy (MLiSSP) (Fig. 4e). The Ref casting solution maintained a high transmission intensity (~90%) with negligible variation, indicating good stability. In comparison, the EOL-CR and EOL-R casting solutions exhibited lower initial transmission intensities (~70% and ~10%, respectively) due to the presence of foulants. For the EOL-R casting solution, the gradual increase in transmission intensity after 12 h suggested partial aggregation and sedimentation of foulants (Fig. 4f)[36]. Despite these differences, the turbiscan stability index (TSI) of both regenerated casting solutions (EOL-R and EOL-CR) remained lower than that of the Ref casting solution within 12 h (Supplementary Fig. 11), indicating better stability of two casting solutions during membrane regeneration[37].

To further investigate the influence of polymer characteristics on the kinetics of the phase inversion process, model structures of low- and high-entangled PVDF chains were constructed to represent the EOL PVDF membrane and commercial PVDF powder, respectively (Fig. 4g). The migration energy barrier of the NMP molecule in the two systems was calculated using first principles. It can be observed that the migration energy barrier of the NMP molecule near high-entangled PVDF chains (32.6 eV) is higher than that near low-entangled PVDF chains (19.3 eV) (Fig. 4h). This finding suggests that both polymer chains and solvent molecules exhibit greater mobility in the casting solution of the low-entanglement system, which facilitates a more uniform rearrangement of polymer chains during solvent outflow, promoting the formation of a well-organized membrane structure[38].

Furthermore, for better understanding the phase inversion behavior, a ternary phase diagram of the EOL-R and Ref system was constructed by determining the cloud point curves via a titration method (see Methods Section)[39]. In the Reference system, the cloud point curve is located closer to the polymer-solvent axis, indicating a larger demixing gap (Fig. 4i). In contrast, the regenerated system exhibits a smaller demixing gap, which can be ascribed to the better compatibility between the polymer and the solvent[40]. The size of the demixing gap reflects the amount of nonsolvent required to induce phase separation. The smaller demixing gap in the regenerated system implies a longer diffusion path is needed to reach the cloud point curve, thereby granting polymer chains more time to rearrange[41]. This extended rearrangement period promotes the formation of a more homogeneous and interconnected polymer network, ultimately contributing to the development of a denser separation layer in the EOL-R membrane[41,42].

## Environmental and economic benefits of membrane regeneration

Two scenarios were devised for assessing the environmental and economic benefits of membrane regeneration strategy (Fig. 5a). With the traditional linear life cycle, EOL membranes were disposed of directly to landfill and replaced with new ones[43]. Conversely, in the membrane regeneration scenario, EOL membranes were redissolved to prepare regenerated membranes. Although the regenerated membranes were prepared from EOL membranes, they exhibited comparable mechanical and thermal stability to the Ref membrane, along with superior separation performance and fouling resistance. Therefore, both membranes were assumed to have a lifespan of 5 years. Using a 60-year operational period for a wastewater treatment plant as the evaluation timeframe, we assessed the economic costs and environmental impacts of the two scenarios.

The total costs for the traditional linear scenario and the membrane regeneration scenario were calculated to be $272.4 m^{-2}$ and $66.1 m^{-2}$, respectively, indicating a 75.7% cost reduction achieved with membrane regeneration (Supplementary Fig. 12a)[44]. This significant cost reduction was primarily attributed to the substantial decrease in expenses associated with the acquisition of new membranes (Supplementary Fig. 12b). Life cycle assessment (LCA) results also indicated reduction in environmental impacts across all studied categories in the membrane regeneration scenario, with $CO_2$-eq emissions showing a reduction of 38.4% (Climate change, CC) (Fig. 5b). These benefits stem primarily from the avoidance of frequent membrane replacement, leading to reductions in polymers usage and other processes involved in manufacturing new membranes, as well as a reduction in EOL membrane disposal[45].

Notably, electricity consumption and solvent usage were the two primary contributors to carbon emissions (Fig. 5c), underscoring the urgent need for sustainable alternatives with a lower carbon footprint and environmental impact. However, the adoption of green solvents faces limitations due to their relatively weak dissolving capacity and harsh preparation conditions[46–48]. Given the superior compatibility and reduced entanglement of EOL membranes with solvent, regenerated membranes using green solvents could be prepared under milder conditions (i.e., reduced electricity consumption). A green solvent, PolarClean®, was used for redissolving the EOL membranes, and the regenerated membrane was successfully fabricated under relatively mild preparation conditions (Supplementary Method 3) compared to those documented in literature[49,50]. The regenerated membranes exhibited water permeance ranging from $585.0 \pm 39.8$ to $1458.3 \pm 56.2 L m^{-2} h^{-1} bar^{-1}$ and a relatively uniform pore size distribution with an average pore size varying from 0.16 μm to 0.92 μm (Supplementary Fig. 13), demonstrating a structure of a skin layer and finger-like pores (Supplementary Fig. 14). Consequently, the adoption of green solvents in membrane regeneration strategy is poised to replace traditional solvents, yielding significant carbon offsets in terms of electricity consumption and solvent usage.

## Discussion

Despite the widely recognized contribution of polymeric membranes to enhancing the sustainability of many industries, the linear life cycle of membrane synthesis, usage, and disposal undermines the sustainability of the technology itself. Regenerating EOL membranes presents an exciting opportunity to enhance the sustainability of membrane applications but is practically viable only if the performance of the regenerated membrane is not substantially compromised. Surprisingly, this work demonstrates that regenerated EOL membranes deliver even higher performance than membranes fabricated with pristine polymer due to a synergy of foulant integration and changes in polymer properties caused by repetitive chemical cleaning during long-term operation. Specifically, foulant integration reduces membrane pore size and increases hydrophilicity, while prolonged exposure to

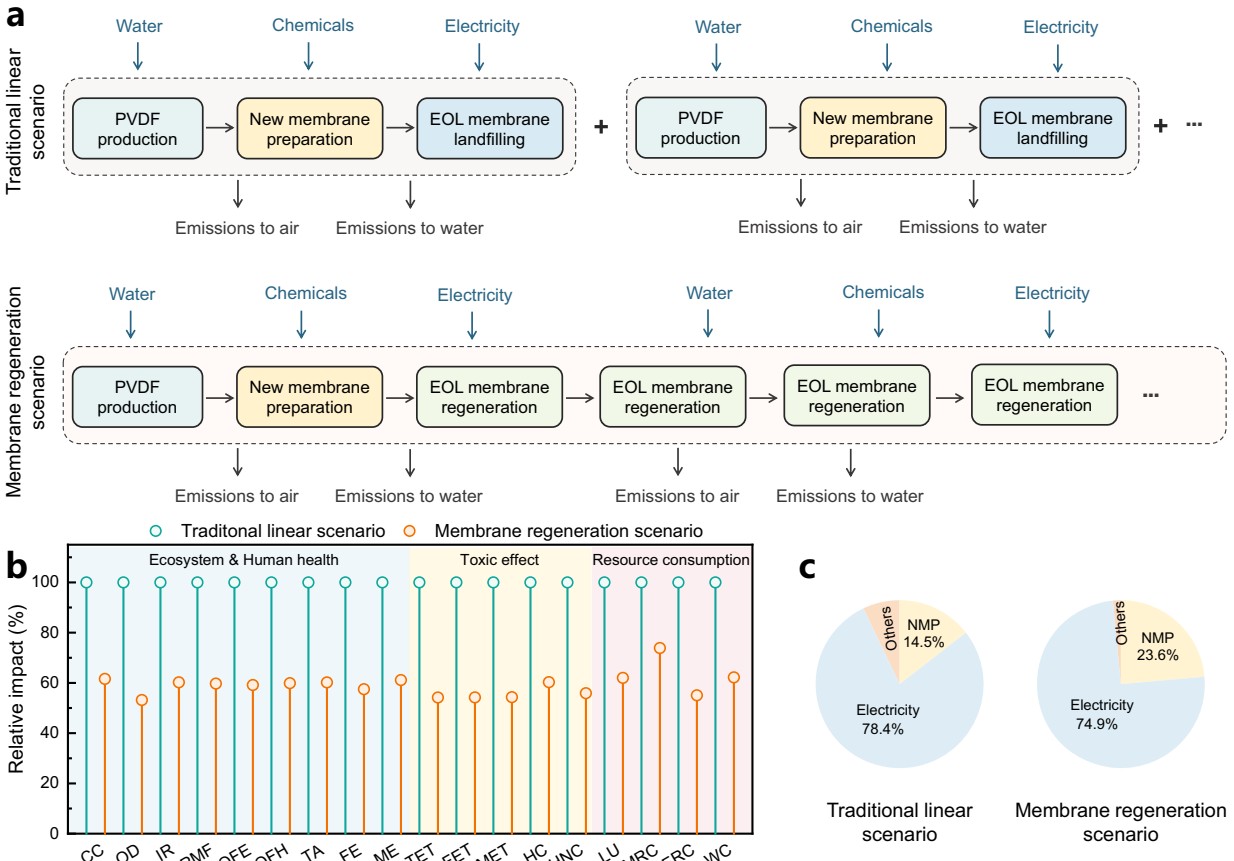

**Fig. 5 | Life cycle impact assessment of two scenarios. a** Schematic diagram of LCA system boundaries describing polymer production, membrane preparation, membrane regeneration and disposal processes in this study. **b** Relative impacts of two scenarios across all impact categories. The first group is related to common ecosystem and human health, including climate change (CC), ozone depletion (OD), ionizing radiation (IR), fine particulate matter formation (FPMF), photochemical oxidant formation: ecosystem quality (POFE), photochemical oxidant formation: human health (POFH), terrestrial acidification (TA), freshwater eutrophication (FE) and marine eutrophication (ME). The second group is related to toxic effect, including terrestrial ecotoxicity (TET), freshwater ecotoxicity (FET), marine ecotoxicity (MET), human toxicity: cancer (HTC) and human toxicity: non-cancer (HTNC). The third group is related to resource and energy consumption, including land use (LU), water use (WU), mineral resource scarcity (MRS) and fossil resource scarcity (FRS). The normalized 100% impacts are impacts of the traditional linear scenario. **c** Contributions of electricity, solvent and others to carbon emissions of each scenario during the process. Source data are provided as a Source Data file.

cleaning agents decreases polymer entanglement, thereby improving compatibility with solvents and foulants. Collectively, these effects lead to higher water permeance, significantly greater contaminant rejection, and lower fouling propensity compared to reference membranes made from pristine polymer powders. Paradoxically, what degrades a membrane during operation can enhance its performance upon regeneration—what kills a membrane makes it even better! This study shows that EOL membrane regeneration is not an inferior sustainable alternative to disposal but rather a transformative approach that enhances membrane performance beyond its original capabilities.

This study employs a solvent-assisted regeneration route, directly reusing the EOL membranes as raw materials. This pattern is fully compatible with existing industrial membrane fabrication lines, providing a scalable and industrially feasible pathway toward the circular membrane industry. By regenerating EOL membranes, this strategy reduces the reliance on raw fossil materials and the energy required for fabricating new membranes, resulting in a 38.4% reduction in $CO_2$-eq emissions and a 75.7% cost saving. Importantly, the process imposes no specific performance requirements for the EOL membranes, and is theoretically applicable to a wide range of polymers (e.g., PVDF, polyether sulfone (PES), and polylactic acid (PLA))[7,17,18].

Although the regenerated membranes exhibited satisfactory separation performance, several challenges remain for industrial

implementation: (1) the inherent diversity of commercial membrane materials (e.g., variations in polymer molecular weight, concentration and additives), demands case-specific optimization of the regeneration process; (2) non-woven support layers are commonly present in commercial membranes, and the recyclability of these fabrics after polymer dissolution remains unclear. To address these challenges, future research should concentrate on evaluating the adaptability of this approach across different membrane materials and establishing standardized regeneration frameworks for diverse membrane materials. In parallel, supportive policies, collection infrastructures, and industry standards will be essential to foster a circular membrane economy and enable large-scale implementation.

## Methods
### General
Materials and chemicals used in this work have been shown in the Supplementary Method 4.

### Real EOL membranes
The EOL hollow fiber PVDF membranes, sourced from a full-scale MBR, were utilized for the preparation of regenerated membranes. The details of new and EOL PVDF membranes are shown in Supplementary Table 2. The PVDF membranes had been in operation for over 6 years,

and the water permeance had significantly deteriorated despite periodic chemical cleaning. Simultaneously, some fibers exhibited breakage due to aging and potential structural damage. The EOL membrane samples were thoroughly rinsed with deionized water and carefully sectioned into 2 ~ 3 cm segments. Subsequently, some of these membrane segments were immersed in a 0.5% NaClO solution in a shaker at 25 °C for 24 h, followed by another 24 h immersion in a 1.5% citric acid solution[21]. These treated membranes were denoted as cleaned membrane segments. Both fouled and cleaned membrane segments were then dried at 80 °C for further use in subsequent procedures.

### Regenerated membrane preparation

Regenerated membranes were fabricated using a phase inversion method via non-solvent induced phase separation, a well-established technique in membrane preparation[51]. Dried membrane segments (9 g, 18 wt%) and NMP (41 g, 82 wt%) were mixed and stirred at 80 °C for 12 h. Then the solution was degassed in an oven at 80 °C for an additional 12 h. The casting solution was uniformly cast onto a nonwoven support with a casting knife gap of 200 μm at a speed of 4 cm s$^{-1}$. The cast films were then immersed in a deionized water bath at room temperature. The resulting membranes, prepared using fouled and cleaned membrane segments, were designated as EOL-R membrane and EOL-CR membrane, respectively. The schematic diagram of the membrane preparation process was shown in Supplementary Fig. 1. Furthermore, the Reference membrane was prepared using the same procedure but with commercial PVDF powder (labeled as Ref membrane). Then the Ref membrane was dissolved using solvent, and a redissolved membrane was prepared through an identical phase inversion process (labeled as Ref-RD membrane).

### Membrane performance evaluation

The water permeance of the membrane was assessed under a pressure of 0.1 MPa using deionized (DI) water as the feed in a dead-end filtration cell (Millipore Cor., U.S.). The cell, with an effective volume of 400 mL and a diameter of 76 mm, was pressurized using a compressed nitrogen cylinder. Prior to testing, the membrane sample was compacted at 150 kPa with DI water until the water permeance reached a stable value. To evaluate the pollutant rejection performance of the membranes, a 0.1 g L$^{-1}$ BSA aqueous solution was used as the feed, and three samples were collected within a 10 min period. The concentration of BSA in the permeate was determined using an ultraviolet spectrophotometer (TU-1810, Beijing Persee Analytics General Instrument Co., Ltd., China).

The fouling-resistance behavior of the membranes was evaluated with a laboratory-scale cross-flow membrane filtration setup. The filtration tests were conducted using model foulant solutions, including 100 mg L$^{-1}$ BSA, 200 mg L$^{-1}$ HA, and 500 mg L$^{-1}$ SA, respectively. All experiments were performed under a pressure of 0.1 MPa, with a cross-flow velocity of 0.2 mL/min. The effective filtration area of the cross-flow cell was 19.0 cm$^2$. The variation in water permeance over time during filtration was recorded (24 h) for each solution.

### Membrane characterization

Scanning electron microscopy (Zeiss Gemini 300, Germany) coupled with energy-dispersive X-ray spectroscopy (EDS) was employed for morphological observation and elemental composition analysis of membrane surfaces. The pore size distribution of membranes was determined using a membrane pore size analyzer (BSD-PB, BSD Instrument) following the standard method of GB/T 32361−2015[52]. X-ray diffraction (XRD, Bruker D8 Advance) analysis was conducted to characterize the crystalline structure of the membrane. Surface functional groups and elemental compositions of membranes were analyzed using attenuated total reflectance Fourier transform infrared spectroscopy (ATR-FTIR, Thermos IN10) and X-ray photoelectron spectroscopy (XPS, Thermo, USA), respectively. The mechanical property of the

membrane, represented by elongation at break, was evaluated through the tensile testing using a universal testing machine (CMT6103, MTS, US)[53]. Thermal stability of the membrane was assessed using a differential scanning calorimeter (DSC, TA Q2000) under a stream of air at a heating rate of 10 °C min$^{-1}$ [54]. An atomic force microscope (AFM, Bruker Dimension Icon, Germany) operating in peak force tapping mode was employed to analyze the membrane surface roughness.

### Effect of foulants incorporation on membrane properties

Based on our previous research, polysaccharide, HA-like substance, protein, Ca, and Si are the main foulants in the EOL membrane from large-scale MBR[21]. To explore the impacts of integrated foulants on regenerated membranes, we then selected BSA as model organic foulant and silicon dioxide (SiO$_2$) as a model inorganic foulant, and blended them into EOL-CR casting solution as membrane additives, respectively. A certain amount of each model foulant, i.e., 0 wt%, 3 wt%, 6 wt%, and 9 wt% (foulant/PVDF, w/w) was added into NMP. An ultrasonic bath at room temperature was conducted to disperse foulant into the solvent for 10 min[55]. The cleaned EOL membrane segments (18 wt%) were added to the solution. Then the casting solution was stirred at 80 °C for 12 h and degassed in an oven at 80 °C for an additional 12 h. The foulant-incorporated membranes were prepared using the same procedure as for the regenerated membrane, labeled as EOL-CR-foulant membranes. Similarly, foulants were added into the Ref membrane, i.e., incorporating BSA and SiO$_2$ with pristine polymer powder, to prepare Ref-foulant membranes.

### Determination of polymer entanglement density

The moduli of casting solutions were measured using a rotational rheometer (Thermo Haake Mars60, Germany) fitted with 60 mm conical plates[28]. The frequency dependence of the dynamic modulus was measured over a frequency range between 0.1 to 628 rad s$^{-1}$ at a constant strain. The relationship between entanglement density ($V_e$), plateau modulus ($G^0_N$) and the maximum of loss modulus ($G''_{max}$) is described by the following equations[27,29]:

$$\frac{G^0_N}{G''_{max}} = 2.303\left[\frac{2}{\pi}\int_{-\infty}^{+\infty}\frac{G''(\omega/\omega_{max})}{G''_{max}}\,\mathrm{d}\,\log(\omega/\omega_{max})\right] = 3.56 \qquad (1)$$

$$V_e \approx \frac{G^0_N}{k_b T} \qquad (2)$$

where $k_b$ is the Boltzmann constant, $T$ is the temperature, and $\omega$ is the frequency.

### Rheology characterization of casting solutions

The dynamic viscosity of casting solutions as a function of shear rate was measured using a rotational rheometer (Thermo Haake Mars60, Germany)[56]. The viscosity was measured over a shear rate range of 0.1 to 1000 s$^{-1}$ at 80 °C. A power law equation was used as a mathematical model for describing the fluid behavior in casting solutions[34].

$$\mu = K\gamma^{n-1} \qquad (3)$$

where $\mu$ is the dynamic viscosity, $\gamma$ is the shear rate, and $K$ and $n$ are flow behavior index parameters. $K$ is an indication of solvent quality, and a smaller $K$ value corresponds to better compatibility between polymer and solvent. The parameter $n$ is determined from the data at high shear rates, typically falls within the range of $0 < n < 1$ for shear thinning solutions.

### MLiSSP characterization of casting solutions

Three casting solutions, prepared with EOL membrane segments, cleaned EOL membrane segments and commercial PVDF powder as

the polymer, respectively, were characterized by multiple light scattering spectroscopy (MLiSSP, Turbiscan Tower, Formulaction, France) (labeled as EOL-R, EOL-CR and Ref casting solutions). MLiSSP automatically monitors the light flux transmitted as a function of height, thereby revealing the real-time destabilization processes in the samples[36]. After preparation, casting solutions were transferred to glass tubes (23 mL of each sample volume) and degassed in an oven at 80 °C for 12 h to avoid interference from bubbles. The tubes were then inserted into the MLiSSP chambers, with the transmission signal collected every 45 mins for 24 h. The stability of samples was characterized using the turbiscan stability index (TSI), which comprehensively reflects processes such as condensation and sedimentation occurring within the system[37]. The TSI value can be determined using the following equation:

$$TSI = \sqrt{\frac{\sum_{i=1}^{n}(x_i - x_T)^2}{n - 1}} \qquad (4)$$

where $x_i$ represents the measured sample transmission intensity at each measurement time point, $x_T$ represents the average transmission intensity, and $n$ represents the number of measurements. A lower TSI value indicates a higher stability of the solution.

### Density functional theory calculations

All structure establishment and spin-polarized calculations based on density functional theory (DFT) were conducted using the DMol3 package, a module integrated into the Materials Studio 2019. The Perdew-Burke-Ernzerhof (PBE) form of the generalized gradient approximation (GGA) and the Semicore Pseudopotential method (DSPP) with double numerical basis sets plus the polarization functional (DNP) were employed[57,58]. In addition a DFT-D correction with the Grimme scheme was applied to include dispersion interactions[59]. The structures of low- and high-entangled PVDF polymer chains were constructed using the unit PVDF. Extensive convergence criteria were established to guarantee computational accuracy. During shape optimization, the self-consistent field (SCF) convergence criterion for each electronic energy was set as $1.0 \times 10^{-6}$ Ha, and the geometry optimization convergence criteria were defined as follows: $1.0 \times 10^{-6}$ Ha for energy, 0.001 Ha Å$^{-1}$ for force, and 0.001 Å for displacement, respectively. Energy barriers of NMP molecules were assessed using linear and quadratic synchronous transit methods combined with conjugated gradient (CG) refinement.

### Cloud point measurements

To determine the cloud point of the two systems, a series of EOL-R and Ref casting solutions with varying polymer concentrations (6 ~ 20 wt%) were prepared and thoroughly mixed (Supplementary Table 3). Each solution was transferred to a glass vessel and maintained at 80 °C. Ultrapure water was then added dropwise using a pipette under continuous mechanical stirring until visible turbidity was observed[39,40]. The amount of water added at the point when the turbidity persisted for at least 30 min was recorded, indicating the onset of liquid-liquid phase separation. Subsequently, the compositions of water, polymer, and solvent at the cloud point for both systems were calculated.

### Life cycle assessment

In order to evaluate the sustainability of membrane regeneration strategy, we conducted a life cycle assessment (LCA) following the ISO 14040 standard. For assessing environmental benefits of membrane regeneration strategy, two scenarios, including traditional linear scenario and membrane regeneration scenario, were devised. A functional unit (FU) of 1 m² membrane was selected for this LCA study. The use phase was excluded from the assessment, given that the target application scenarios of membranes are consistent[43,45]. Process inventory data were obtained from experiments, engineering experience, vendors, and literature (Supplementary Table 4 and Table 5). The background processes, such as energy supply and material extraction, were obtained from the Ecoinvent v3.8 cutoff database. Life cycle impact assessment was conducted using the widely-used ReCiPe method with midpoint indicators, which provides results of higher certainty for impact categories. Specifically, all the impact categories were selected and divided into three groups: common ecosystem and human health impact categories, toxic effect impact categories and resource and energy consumption impact categories. Details of the impact categories are provided in Supplementary Table 6.

### Economic analysis

To ensure consistency between environmental and economic analyses, identical system boundaries and scenarios were employed for the LCA study and the economic analysis. A detailed cost calculation spreadsheet is provided in Supplementary Table 7, encompassing market prices of new commercial membranes, disposal costs of EOL membranes, and expenses related to the regeneration process. In addition, expenses associated with membrane regeneration, such as solvent, electricity, and labor costs, were considered.

## Data availability

The data supporting the findings of the study are included in the main text and supplementary information file. The atomic coordinates of configurations used in this study are provided in Supplementary Data 1, 2, 3. Source data are provided in this paper.

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

## Acknowledgements

This work was supported by the National Natural Science Foundation of China, awarded to R.B.D. (52522006 & 52470091) and Z.W.W. (52430001), and the Shanghai Municipal Science and Technology Commission International Cooperation Project to R.B.D. (24230712000).

## Author contributions

C.X.T., R.B.D., and Z.W.W. conceived the idea and designed the research. C.X.T. performed the experiment, including membrane preparation, characterization, performance test and mechanism verification. J.S.X.C. performed a life cycle assessment. C.X.T., R.B.D., and Z.W.W. contributed to the writing of the manuscript. Z.W.Q. provided suggestions for membrane preparation. S.H.L. provided constructive suggestions for results and discussion.

## Competing interests

The authors declare no competing interests.
