## [Transparent Peer Review file · Nature Communications]

Regenerating End-of-Life Membranes for Enhanced Sustainability and Unexpected Performance

Corresponding Author: Professor Ruobin Dai

Version 0:

Reviewer comments:

Reviewer #2

(Remarks to the Author)

The authors have made detailed and targeted revisions to the manuscript in response to comments. In the revised version, the applicability of the membrane regeneration strategy, the generalizability of foulant effects, and the mechanisms underlying the performance enhancement of regenerated membranes have all been further validated and discussed through experiments. This study provides a universal technical framework for the reuse of end-of-life membranes, offering valuable insights for the sustainable development of membrane technology. I believe the manuscript is now suitable for publication and recommend acceptance. Nevertheless, I would like to offer a few minor suggestions that may further improve the manuscript:

1. When discussing the stability of the casting solution, if I understand correctly, the key point is that the foulants do not adversely affect the stability of the casting solution within a certain period. Therefore, some descriptions regarding casting solution properties can be simplified to emphasize this main conclusion.
2. The manuscript repeatedly uses the transition word "furthermore". While the overall writing is clear, I recommend varying the transitional phrases to improve the flow.
3. The figures consist of multiple subfigures and contain rich, detailed data. It is recommended to enlarge the font size within the figures to enhance clarity and legibility.
4. When presenting the DFT-calculated migration energy barrier (Fig. 4h), consider reporting the absolute values (in kJ/mol or eV) in the main text, not only in the figure. This will help readers better quantify and interpret the results.
5. Membrane regeneration represents a promising approach with strong potential for future industrial implementation. To facilitate its scale-up, collaborative efforts are needed to establish supportive policies, recovery infrastructure, and industrial standards that enable a circular membrane economy. A brief discussion of this aspect could be added to the manuscript.
6. The final Discussion section appears somewhat lengthy, possibly due to efforts to fully address reviewer comments. To enhance clarity and focus, I suggest condensing this section by emphasizing the most critical mechanistic insights and future perspectives.

Reviewer #3

(Remarks to the Author)

The authors have provided thoughtful responses to the questions, but the answers regarding innovation are still not convincing. As noted by the reviewer, there have been many similar studies. Therefore, it is not recommended for publication in the Journal of Nature Communications.

Version 1:

Reviewer comments:

Reviewer #2

(Remarks to the Author)

The work can be accepted now.

Response to the reviewers' comments and List of Changes Made in the Manuscript

Reviewer #2

The authors have made detailed and targeted revisions to the manuscript in response to comments. In the revised version, the applicability of the membrane regeneration strategy, the generalizability of foulant effects, and the mechanisms underlying the performance enhancement of regenerated membranes have all been further validated and discussed through experiments. This study provides a universal technical framework for the reuse of end-of-life membranes, offering valuable insights for the sustainable development of membrane technology. I believe the manuscript is now suitable for publication and recommend acceptance. Nevertheless, I would like to offer a few minor suggestions that may further improve the manuscript:

Our response: We thank the reviewer for the positive comment. We have carefully revised our manuscript based on the following suggestions, with the point-to-point responses listed below.

1. *When discussing the stability of the casting solution, if I understand correctly, the key point is that the foulants do not adversely affect the stability of the casting solution within a certain period. Therefore, some descriptions regarding casting solution properties can be simplified to emphasize this main conclusion.*

Our response: Thanks for the comment. We have simplified the descriptions regarding casting solution properties and emphasized the better stability of two casting solutions (EOL-R and EOL-CR) during membrane regeneration. Corresponding discussion has been shown in the revised manuscript.

Lines 297-305 of the revised manuscript:

“The Ref casting solution maintained a high transmission intensity (~90%) with negligible variation, indicating good stability. In comparison, the EOL-CR and EOL-R casting solutions exhibited lower initial transmission intensities (~70% and ~10%, respectively) due

to the presence of foulants. For the EOL-R casting solution, the gradual increase in transmission intensity after 12 h suggested partial aggregation and sedimentation of foulants (Fig. 4f)³⁶. Despite these differences, the turbiscan stability index (TSI) of both regenerated casting solutions (EOL-R and EOL-CR) remained lower than that of the Ref casting solution within 12 h (Extended Data Fig. 4a), indicating better stability of two casting solutions during membrane regeneration³⁷.”

2. The manuscript repeatedly uses the transition word “furthermore”. While the overall writing is clear, I recommend varying the transitional phrases to improve the flow.

Our response: Thank you for the comment. To improve the readability and logical flow, we have revised the manuscript accordingly by varying the transitional words. Specifically, “furthermore” has been replaced with more appropriate alternatives such as “in addition”, “moreover” and “importantly”. Corresponding description has been shown in the revised manuscript.

Lines 159-167 of the revised manuscript:

“In addition, the regenerated membranes displayed enhanced hydrophilicity as evidenced by a smaller water contact angle (Fig. 2j). The improved surface hydrophilicity is likely attributable to the presence of the incorporated foulants, which is supported by the lower F 1s and higher O 1s content observed in the regenerated membrane as compared to the Ref membrane (Supplementary Table 1). Moreover, the regenerated membrane demonstrated a smoother surface ($R_a = 41.5 \pm 9.7$ nm for EOL-R and 38.2 ± 5.6 nm for EOL-CR) compared to the Ref membrane ($R_a = 64.6 \pm 6.9$ nm) (Supplementary Fig. 1e). These changes in surface characteristics play important roles in enhancing the fouling resistance of the regenerated membrane.”

Lines 398-400 of the revised manuscript:

“Importantly, the process imposes no specific performance requirements for the EOL membranes, and is theoretically applicable to a wide range of polymers (e.g., PVDF, polyether sulfone (PES), and polylactic acid (PLA))^{7,17,18}.”

3. The figures consist of multiple subfigures and contain rich, detailed data. It is recommended to enlarge the font size within the figures to enhance clarity and legibility.

Our response: Thank you for the suggestion. We have enlarged the font size within the figures to enhance clarity and legibility. Please see Fig. 2 and Fig. 5 in the revised manuscript:

Fig. 3. The effect of foulant incorporation on membrane performance. The membranes blended with foulant at 0, 3, 6, and 9 wt% was labeled as EOL-CR-foulant-0, 1, 2, 3, respectively. The performance and characterization of EOL-CR-foulant membranes: **a**, The water permeance and rejection rate. **b**, Pore size distribution. **c**, Water contact angle ($n = 5$). The dashed line and gray bar represent the average BSA rejection and its error bar of the EOL-CR membrane. **d**, The water permeance and rejection rate of Ref-foulant membranes. **e**, Performance enhancement ratio of EOL-CR-foulant membranes compared to Ref-foulant membranes: water permeance and rejection rate. **The element content in cross section of foulant-incorporated membranes:** **f**, O element content in cross section of EOL-CR-BSA and Ref-BSA membranes. **g**, Si element content in cross section of EOL-CR-SiO₂ and Ref-SiO₂ membranes. Error bars in **a**, **d**, **e**, **f**, **g** represent the s.d. ($n = 3$) and data are presented as mean values \pm s.d.

Fig. 5. Life cycle impact assessment of two scenarios. a, Schematic diagram of LCA system boundaries describing polymer production, membrane preparation, membrane regeneration and disposal processes in this study. **b**, Relative impacts of two scenarios across all impact categories. The normalized 100% impacts are impacts of traditional linear scenario. **c**, Contributions of electricity, solvent and others to carbon emissions of each scenario during the process.

4. When presenting the DFT-calculated migration energy barrier (Fig. 4h), consider reporting the absolute values (in kJ/mol or eV) in the main text, not only in the figure. This will help readers better quantify and interpret the results.

Our response: Thank you for the comment. To improve clarity and facilitate quantitative interpretation, the absolute values of the DFT-calculated migration energy barriers (in eV) have been added to the revised manuscript.

Lines 310-315 of the revised manuscript:

“It can be observed that the migration energy barrier of NMP molecule near high-entangled PVDF chains (32.6 eV) is higher than that near low-entangled PVDF chains (19.3 eV) (Fig. 4g, h). This finding suggests that both polymer chains and solvent molecules exhibit greater mobility in the casting solution of low-entanglement system, which facilitates a more uniform rearrangement of polymer chains during solvent outflow, promoting the formation of a well-organized membrane structure³⁸.”

5. *Membrane regeneration represents a promising approach with strong potential for future industrial implementation. To facilitate its scale-up, collaborative efforts are needed to establish supportive policies, recovery infrastructure, and industrial standards that enable a circular membrane economy. A brief discussion of this aspect could be added to the manuscript.*

Our response: Thank you for the comment. In response, we have added a discussion on the policy, infrastructure, and standardization requirements for promoting a circular membrane economy in the revised Discussion section.

Lines 407-412 of the revised manuscript:

“To address these challenges, future research should concentrate on evaluating the adaptability of this approach across different membrane materials and establishing standardized regeneration frameworks for diverse membrane materials. In parallel, supportive policies, collection infrastructures, and industry standards will be essential to foster a circular membrane economy and enable large-scale implementation.”

6. *The final Discussion section appears somewhat lengthy, possibly due to efforts to fully address reviewer comments. To enhance clarity and focus, I suggest condensing this section by emphasizing the most critical mechanistic insights and future perspectives.*

Our response: We thank the reviewer for this valuable suggestion. In the revised manuscript, the Discussion section has been condensed and reorganized into three focused parts for improved clarity and logical flow. The first part now concentrates on the unexpected enhanced performance of regenerated membranes and the

underlying mechanisms. The second part emphasizes the applicability and scalability of the regeneration strategy, describing its compatibility with industrial fabrication processes and its potential for various polymer materials. The third part outlines future research directions.

Lines 376-411 in the revised manuscript.

“Despite the widely recognized contribution of polymeric membranes to enhancing the sustainability of many industries, the linear life cycle of membrane synthesis, usage, and disposal undermines the sustainability of the technology itself. Regenerating EOL membranes presents an exciting opportunity to enhance the sustainability of membrane applications but is practically viable only if the performance of the regenerated membrane is not substantially compromised. Surprisingly, this work demonstrates that regenerated EOL membranes deliver even higher performance than membranes fabricated with pristine polymer due to a synergy of foulant integration and changes in polymer properties caused by repetitive chemical cleaning during long-term operation. Specifically, foulant integration reduces membrane pore size and increases hydrophilicity, while prolonged exposure to cleaning agents decreases polymer entanglement, thereby improving compatibility with solvents and foulants. Collectively, these effects lead to higher water permeance, significantly greater contaminant rejection, and lower fouling propensity compared to reference membranes made from pristine polymer powders. Paradoxically, what degrades a membrane during operation can enhance its performance upon regeneration—what kills a membrane makes it even better! This study shows that EOL membrane regeneration is not an inferior sustainable alternative to disposal but rather a transformative approach that enhances membrane performance beyond its original capabilities.

This study employs a solvent-assisted regeneration route, directly reusing the EOL membranes as raw materials. This pattern is fully compatible with existing industrial membrane fabrication lines, providing a scalable and industrially feasible pathway toward circular membrane industry. By regenerating EOL membranes, this strategy reduces the reliance on raw fossil materials and the energy required for fabricating new membranes, resulting in a 38.4% reduction in CO₂-eq emissions and a 75.7% cost saving. Importantly, the process imposes no specific performance requirements for the EOL membranes, and is

theoretically applicable to a wide range of polymers (e.g., PVDF, polyether sulfone (PES), and polylactic acid (PLA))^{7,17,18}.”

Although the regenerated membranes exhibited satisfactory separation performance, several challenges remain for industrial implementation: (1) the inherent diversity of commercial membrane materials (e.g., variations in polymer molecular weight, concentration and additives), demands case-specific optimization of the regeneration process; (2) non-woven support layers are commonly present in commercial membranes, and the recyclability of these fabrics after polymer dissolution remains unclear. To address these challenges, future research should concentrate on evaluating the adaptability of this approach across different membrane materials and establishing standardized regeneration frameworks for diverse membrane materials. In parallel, supportive policies, collection infrastructures, and industry standards will be essential to foster a circular membrane economy and enable large-scale implementation.”

Reviewer #3

The authors have provided thoughtful responses to the questions, but the answers regarding innovation are still not convincing. As noted by the reviewer, there have been many similar studies. Therefore, it is not recommended for publication in the Journal of Nature Communications.

Our response: We sincerely thank the reviewer for the additional feedback and for emphasizing the importance of clearly articulating our innovation. We understand the concern that some studies have explored waste polymer recycling by different methods or membrane downcycling by removing the top layer of RO membranes. However, we emphasize that our study introduces a fundamentally different regeneration paradigm, a direct membrane-to-membrane circular route that not only reclaims waste membranes but also enhances their separation performance beyond the original pristine membranes. To clarify this distinction, we provide a comparative summary (Table 1), highlighting the key differences between the representative studies cited by the reviewer and our work.

Table 1. Comparison between representative waste recycling studies and this work

Initial material	Method	Final product	Mechanism /Focus	Distinct difference from this work	Study
Waste plastics (PE, PET, PS)	Chemical depolymerization via electrothermal technologies	Chemicals, fuels, and carbon materials	General chemical recycling of plastics	Focuses on plastic waste valorization, not functional membrane regeneration	Luo et al., 2025
Waste PET bottles	Fiber extraction and membrane fabrication	Biofuel separation membrane	Converting plastics into membranes	Converts bottle waste into membranes, not membrane to membrane	Saeed et al., 2025
Waste polymer membranes	Solvent-catalyzed conversion	Porous carbon adsorbents	Adsorbent preparation from polymer waste	Loses separation function of membranes; no regeneration of original use	Santhosh et al., 2024a
Waste polymer membranes	Deep eutectic solvent-assisted treatment	Energy storage materials	Energy application of waste polymers	Converts membranes into unrelated materials	Santhosh et al., 2024b

End-of-life RO membranes	NaClO treatment	Loose NF or UF membranes	Degradation of polyamide layer	Converts membranes into lower-precision membranes	Wang et . al. (2024)
End-of-life UF membranes	Dissolution–regeneration (direct recycling)	Regenerated membranes with enhanced performance	Synergistic enhancement via foulant integration and polymer entanglement reduction	/	This work

The distinctive innovations of our study compared with prior works have been summarized below:

1. Conceptual innovation: a true membrane-to-membrane circular route

Previous studies mainly focus on the conversion of waste membranes into other materials (adsorbents, carbon, or energy storage materials) or the conversion of plastic waste into membrane materials. In contrast, our study establishes a closed-loop recycling process in which end-of-life (EOL) membranes are regenerated into functional membranes of the same type. This approach preserves the original separation function while achieving improved performance, thereby realizing true circularity within the membrane industry.

2. Mechanistic innovation: degradation-driven enhancement

This work reveals a counterintuitive mechanism where fouling and cleaning during long-term operation improves the performance of regenerated membranes. Foulant integration reduces pore size and enhances hydrophilicity, while cleaning-induced polymer disentanglement improves polymer-solvent compatibility. The combined effects result in higher water permeance and greater rejection efficiency than those of membranes fabricated from pristine polymers. Such a degradation-induced performance enhancement has not been reported in previous recycling studies.

3. Technical innovation: scalable regeneration process

Our regeneration route avoids chemical depolymerization or monomer synthesis and instead uses a solvent-assisted regeneration process that is fully compatible with existing industrial membrane fabrication lines. The process directly reuses

the existing polymer matrix and incorporated foulants as raw materials. Its compatibility with existing production lines makes it a practical and scalable solution. Previous studies mainly emphasized material valorization into new products, whereas this work establishes a practical closed-loop regeneration route within the same membrane application.

4. Industrial and environmental significance

This work provides a practical solution to the challenge of end-of-life membrane disposal by establishing a circular and sustainable route within the membrane industry. It creates a closed-loop supply chain that reintroduces used membranes into the same application domain, reducing dependence on virgin polymer resources and lowering environmental impact (achieving 38.4% lower CO₂-equivalent emissions and 75.7% cost reduction compared with conventional fabrication). Beyond scientific novelty, this study demonstrates substantial potential for industrial transformation and contributes to the realization of a circular economy in membrane technology.

These key aspects underscore the unique conceptual, scientific, technological, and industrial advances of our study, distinguishing it from the works cited by the reviewer. The corresponding discussion has been added in the revised manuscript.

Lines 393-398 in the revised manuscript:

“This study employs a solvent-assisted regeneration route, directly reusing the EOL membranes as raw materials. This pattern is fully compatible with existing industrial membrane fabrication lines, providing a scalable and industrially feasible pathway toward circular membrane industry. By regenerating EOL membranes, this strategy reduces the reliance on raw fossil materials and the energy required for fabricating new membranes, resulting in a 38.4% reduction in CO₂-eq emissions and a 75.7% cost saving.”